# Robust Correction of Sampling Bias using Cumulative Distribution Functions

**Bijan Mazaheri**
Department of Computing and Mathematical Sciences,
California Institute of Technology,
Pasadena, CA 91125
bmazaher@caltech.edu

**Siddharth Jain**
Department of Electrical Engineering,
California Institute of Technology, Pasadena, CA, 91125
sidjain@caltech.edu

**Jehoshua Bruck**
Department of Electrical Engineering,
California Institute of Technology, Pasadena, CA, 91125
bruck@caltech.edu

## Abstract

Varying domains and biased datasets can lead to differences between the training and the target distributions, known as covariate shift. Current approaches for alleviating this often rely on estimating the ratio of training and target probability density functions. These techniques require parameter tuning and can be unstable across different datasets. We present a new method for handling covariate shift using the empirical cumulative distribution function estimates of the target distribution by a rigorous generalization of a recent idea proposed by Vapnik and Izmailov. Further, we show experimentally that our method is more robust in its predictions, is not reliant on parameter tuning and shows similar classification performance compared to the current state-of-the-art techniques on synthetic and real datasets.

## 1   Introduction

Traditional machine learning approaches assume that training data and target data are drawn from the same distribution. Under this assumption, finding the model which minimizes the error on the training dataset also minimizes the expected error in the target domain.

$$\arg\min_f \mathbb{E}_{x \in X_{\text{target}}}[L(f(x), y)] \approx \arg\min_f L(f(X_{\text{train}}), y_{\text{train}}) \tag{1}$$

In practice, sampling bias can lead to a breakdown of the assumption in Eq.1. Certain populations may be more likely to answer a survey or seek a test, leading to over-representation in these demographics. In addition, researchers may wish to re-purpose data outside of an easily sampled domain. For example, a cancer researcher may use European data with a target domain of North America. In these situations, the errors of over-represented regions exert excessive influence on the selection of the model when minimizing training error (see Figure 1a).

Formally, let the joint distribution of input $x$ and output $y$ on the training and target distributions be given by $p(x, y)$ and $q(x, y)$ respectively. We note that $p(x, y) = p(x)p(y|x)$ and $q(x, y) =$

$q(x)q(y|x)$. This paper concerns itself with handling a shift in the input probability distribution $(p(x) \neq q(x))$ with an unchanged output function $(p(y|x) = q(y|x))$. This specific type of mismatch between $p(x, y)$ and $q(x, y)$ is known as *covariate shift*. We will consider cases where the the target distribution $q(x)$ can be sampled from, but not labeled. For example, the age distribution of a population may be known, but we may only have access to labeled data from surveys from one neighborhood.

Past approaches for covariate shift have suggested re-weighting the loss incurred by each training point using estimates of the importance weights. More specifically, [11] suggests handling differences in the distribution by looking at the ratio of probability densities of the covariates in the training $(p(x))$ and target distribution $(q(x))$. The weight for each datapoint in the weighted loss function is given by $w_1(x) = \frac{q(x)}{p(x)}$. By weighing loss at a point according to this scheme, the apparent density of covariate data shifts from $p$ to $q$. Hence, minimizing this new weighted loss function $L_w(f(x), y)$ on the *training* data minimizes the expected loss $L(f(x), y)$ under the *target* distribution on $x$. [14] and [19] modify this framework by regularizing the ratio between density functions. [14] suggests $w_2(x) = (\frac{q(x)}{p(x)})^\tau$ using a regularization parameter, $\tau$. This parameter is tuned using cross validation on the training data, which is weighted according to importance sampling [10].

A key shortcoming of these methods is that they require estimating the probability density function of both the training and target data using density estimation techniques like kernel density estimation (KDE) [2]. These methods for covariate shift are extremely sensitive to the bandwidth ($h$) used in KDE (see Figure 1b). The tuning of this bandwidth, as well as the regularization parameters suggested in [14] require cross validation, which limits the data available for training. Less well-behaved distributions may also differ in optimal bandwidth for different subsets of their domain, further complicating the problem [16].

[15, 6, 19] propose bypassing the KDE step by solving for importance weights that minimize KL-divergence [15] (KLIEP) or unconstrained least squares [6] (uLSIF) or the extension of uLSIF to relative uLSIF (RuLSIF) [19]. Unfortunately, these methods still rely on parameter tuning when choosing the basis functions used in the algorithm, which gives rise to the same limitations. A different method based on moment-matching is suggested in [5], but again requires tuning of parameters for solving the optimization problem of interest. [8] uses a minimax estimation formulation to improve robustness to a worst case scenario. While this method does not require explicit parameter tuning, it still requires an arbitrary selection of features, such as moments, that characterize the source distribution.

While previous works focus on the *accuracy* of classification, we additionally concern ourselves with accurately and stably learning the conditional probability of the output. Previous methods for handling covariate shift suffer from *instability* (non-robustness) in prediction of the target probability function (see Figure 1c). This instability may not affect classification accuracy severely, but it poses a severe limitation on the interpretability of the results.

Recently, [17] put forth a framework for fitting the conditional probability of labels. A key advantage of this framework is that it is based on empirically estimating the cumulative distribution function from data, rather than the probability density function. The cumulative distribution function smooths granularities that exist in empirical data, which makes this method more *robust or stable* in its estimation of conditional probability. In this paper we adapt [17]'s framework to the covariate shift problem. While [17]'s derivation assumes a uniform distribution, we define a loss function that minimizes the expected error over an estimate of the *target* distribution (see Figure 4). As our method relies solely on the empirical estimate of the cumulative distribution function of the target distribution, it does not require any parameter tuning which is a significant advantage over the other widely known methods.

## 1.1 Contributions

- We calculate the $V$-matrix introduced in [17] using an empirical estimate of the cumulative distribution of the target distribution, showing that the estimate of the $V$-matrix obtained using empirical cumulative distribution is the minimum variance unbiased estimator (MVUE) of the true $V$ (Lemma 3.1). We also provide tight rate of convergence theorems for the empirical estimate of $V$ to the true $V$ for one dimensional (Theorem 3.2) and higher dimensional data (Theorem 3.3).

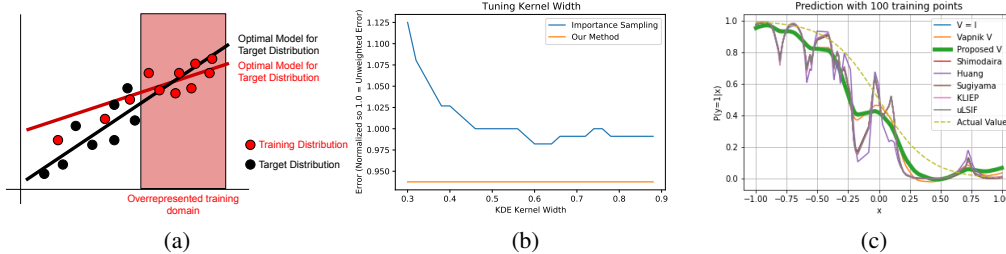

(a)　　　　　　　　　　(b)　　　　　　　　　　(c)

Figure 1: (a) Illustrating covariate shift: The training data (red) is concentrated more in the red rectangular box while the target data (black, drawn from the target distribution) has samples outside the red box. (b) A demonstration of how the choice of bandwidth during the covariate shift can lead to wildly different performance in traditional importance weighting [11]. This represents one run of Experiment 4 for the ringnorm dataset. (c) Previous methods are not stable when giving probability estimates. The smoother prediction curve of our proposed method compared to other widely known covariate shift methods demonstrating the *robustness* of the proposed method. For details of parameters used, refer to section 3.3.

- We experimentally compare the proposed covariate shift method against the widely known covariate shift methods [11, 5, 14, 15, 6] using the Support Vector Machine (SVM) algorithm and show more *robust* predictions and *reduced $L2 - prediction$* error on a synthetic dataset.

- We conduct experiments on real datasets [18, 20, 9, 13, 21] and show comparable performance to other widely known covariate shift methods [11, 5, 14, 15, 6]. For most other methods, performance relies heavily on tuning the right parameters. In some situations, if the parameters were not correctly tuned, their performance was much worse than the unweighted classifier. Methods for automatically tuning parameters like [15, 6] sometimes fail to find these optimal states, also performing worse than unweighted classifiers. On the other hand, since our method does not require any tuning of parameters, it rarely performs significantly worse than an unweighted classifier. Our method outperforms all other tested methods in certain experiments and shows consistent good performance in different settings.

The rest of the paper is organized as follows. In Section 2, we introduce the key details of the framework described in [17]. In section 3, we present our method, algorithms and experiments for covariate shift under [17]'s framework. In section 4, we conclude the paper.

## 2  Preliminaries

Let $z = (z^1, z^2, \cdots, z^n) \in \mathbb{R}^n$, $x \in \mathbb{R}$ and $S \subseteq \mathbb{R}$. We define threshold and indicator functions:

$$\theta(z). = \begin{cases} 1, & \text{if } z^i \geq 0 \,\forall\, i \\ 0, & \text{otherwise.} \end{cases} \qquad\qquad \mathcal{I}_{x \in S} = \begin{cases} 1, & \text{if } x \in S \\ 0, & \text{otherwise.} \end{cases}$$

### 2.1  Fredholm Integral Equation

The conditional probability distribution function $p(y = 1 \mid x)$ is defined as the solution $f(x)$ of the following Fredholm equation:

$$\underbrace{\int_{\mathbb{R}^n} \theta(x - x') f(x') dP(x')}_{F_1(x)} = \underbrace{P(y = 1, x)}_{F_2(x)} \tag{2}$$

We note that this definition of the conditional probability function does not require the existence of a density function. In order to estimate $f(x)$, we need to find the solution of Equation (2) above using the training dataset $(x_1, y_1), (x_2, y_2), \cdots, (x_N, y_N)$. [17] estimates $f(x)$ using the empirical distribution approximations of $P(x)$ and $P(y = 1, x)$ which are given by $P_N(x) = \frac{1}{N} \sum_{i=1}^N \theta(x - x_i)$

and $P_N(1, x) = \frac{1}{N} \sum_{i=1}^{N} y_i \theta(x - x_i)$. Substituting these empirical distributions $P_N(x)$ and $P_N(1, x)$ in Eq. (2) we obtain,

$$F_1 = \frac{1}{N} \sum_{i=1}^{N} f(x_i)\theta(x - x_i) \qquad\qquad F_2 = \frac{1}{N} \sum_{i=1}^{N} y_i\theta(x - x_i) \qquad (3)$$

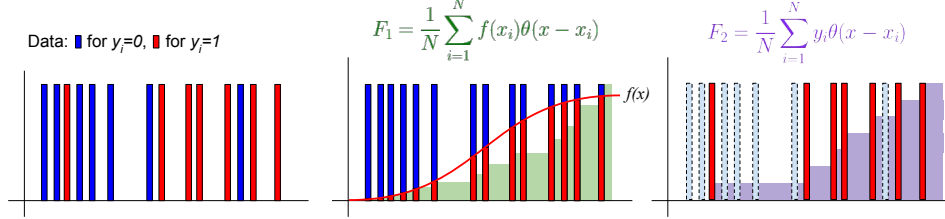

Figure 2: A visualization of $F_1$ and $F_2$, which we seek to make similar in $L2$ norm. In $F_1$, the CDF increases at all locations of data, incrementing according to the value of $f(x)$. In $F_2$, the CDF increases by equal increments for all points with $y_i = 1$.

## 2.2 Loss Function

The solution of equation 2 is found by minimizing the following loss function $L(f)$:

$$L(f) = \rho^2(F_1, F_2) + \gamma W(f) \qquad (4)$$

$\rho^2(F_1, F_2)$ is a measure of distance between $F_1$ and $F_2$, $W(f)$ is the regularizing function and with regularization constant $\gamma$. Under the $L2$ metric $\rho^2(F_1, F_2)$ is given by:

$$\rho^2(F_1, F_2) = \int (F_1(x) - F_2(x))^2 \phi(x) d\mu(x) \qquad (5)$$

where $\phi(x) \geq 0$ is a weight function that we will leave as constant. $\mu(x)$ is a probability measure defined on domain $\mathcal{D}$ consisting of $x \in \mathbb{R}^n$. Note that $\mu(x)$ is a cumulative distribution function. Plugging in the empirical estimates for $F_1$ and $F_2$ from Eq. 3 gives us

$$\rho^2 = \frac{1}{N^2} \int_{\mathbb{R}^n} \left( \sum_{i=1}^{N} \theta(x - x_i)f(x_i) - \sum_{j=1}^{N} y_j\theta(x - x_j) \right)^2 \phi(x)d\mu(x) \qquad (6)$$

Eq. (6) simplifies to

$$\rho^2 = \frac{1}{N^2} \sum_{i,j=1}^{N} (f(x_i) - y_i)(f(x_j) - y_j)V(i, j) \qquad (7)$$

Here $V$ denotes a $n \times n$ symmetric matrix with $V(i, j)$ given by equation (8). Intuitively, this matrix factors into account the mutual positions of the points $x_i$ and $x_j$. Note here if $V$ is an identity matrix, we recover the classical $L2$ loss function.

## 2.3 V-matrix

The $V$-matrix is given by the following expression:

$$V(i, j) = \int_{\mathbb{R}^n} \theta(x - x_i)\theta(x - x_j)\phi(x)d\mu(x). \qquad (8)$$

if $\mu(x) = \prod_{k=1}^{n} \mu_k(x^k)$, Eq. (8) can be simplified when $\phi(x) = 1$ and is given by the following lemma (proof in Supplementary material)

**Lemma 2.1.** *For* $\phi(x) = 1$ *and* $\mu(x) = \prod_{k=1}^{n} \mu_k(x^k)$, *then* $V(i, j) = \prod_{k=1}^{n} \left[ 1 - \mu_k(max(x_i^{k-}, x_j^{k-})) \right]$.

**Note:** If $\mu_k(.)$ is left-continuous at $x^k$, then $\mu_k(x^{k-}) = \mu_k(x^k)$.

We state below examples of V-matrix calculated using Lemma 2.1 for two different choices of $\mu(x)$ with $\phi(x) = 1$.

**Example 2.1.** $\phi(x) = 1, n = 1, x \sim \mathcal{N}(0,1)$, then $V(i,j) = \frac{1}{2}\left[1 - erf\left(\frac{max(x_i, x_j)}{\sqrt{2}}\right)\right]$.

**Example 2.2.** $\phi(x) = 1, x^k \sim U[-c^k, c^k], \ 1 \le k \le n$ (U denotes uniform distribution), then $V(i,j) = \frac{1}{\prod_{k=1}^{n} 2c^k} \prod_{k=1}^{n}[c^k - max(x_i^k, x_j^k)]$.

Note that $V$-matrix does not depend on labels $y_i$. [17] showed the application of the $V$-matrix in example 2.2 to Support Vector Machines [1], termed V-SVM, which we will now discuss.

## 2.4 V-SVM

By assuming that $f(x) = \sum_{i=1}^{N} \alpha_i K(x_i, x) + c$ and $W(f) = \sum_{i,j=1}^{N} \alpha_i \alpha_j K(x_i, x_j)$ for a given Kernel function $K(.,.)$, [17] derived a closed form solution for $f(x)$ given the loss function $L(f)$ in Eq. (4) where the term $\rho^2(.,.)$ is given by Eq. (7). Before stating the solution for $f(x)$, we define the following notation:

$A = (\alpha_1, \alpha_2, \cdots, \alpha_N)^T$, $\mathcal{K}(x) = (K(x_1, x), K(x_2, x), \cdots, K(x_N, x))^T$, $N \times N$ dimensional matrix $K$ with $K_{ij} = K(x_i, x_j)$ being the $ij^{th}$ entry, $Y = (y_1, y_2, \cdots, y_N)^T$, $N$- dimensional vector $1_N = (1, 1, \cdots, 1)^T$, $I$ being the $N \times N$ identity matrix.

Note under this notation $f(x)$ can be rewritten as $f(x) = A^T \mathcal{K}(x) + c$. [17] derived the closed form solution for $A$ and $c$ which is given by $A = A_b - cA_c$, where $A_b = (VK + \gamma I)^{-1} VY$, $A_c = (VK + \gamma I)^{-1} V 1_N$, $c = \frac{1_N^T V(KA_b - Y)}{1_N^T V(KA_c - 1_N)}$. Note here if $V = I_N$, we recover the solution for the classical SVM with L2-error. In the next section, we consider handling covariate shift in this framework by deriving the V-matrix using samples from the target distribution. As seen in Eq. (8), the V-matrix is independent of the labels, hence unlabeled test data information is sufficient for this procedure.

# 3   Results: Using Test Data from the Target Distribution in Learning

**Algorithm 1** Covariate Shift Classification 1

> **Input:** $D = (x_1, y_1), (x_2, y_2), \cdots, (x_N, y_N),$
> $\quad\quad T = t_1, t_2, \cdots, t_M.$
> **Output:** $p(y = 1|x)$

1: **procedure** COMPUTEV$(D, T)$
2: $\quad V \leftarrow 0^{N \times N}$
3: $\quad$ **for** $i, j \le N$ **do**
4: $\quad\quad$ **for** $q \le M$ **do**
5: $\quad\quad\quad$ **if** $x_i \preceq t_q$ and $x_j \preceq t_q$ **then**
6: $\quad\quad\quad\quad V_{ij} \leftarrow V_{ij} + 1$
7: $\quad\quad V_{ij} \leftarrow \frac{V_{ij}}{M}$
$\quad\quad$ **return** V

1: $V \leftarrow$ COMPUTEV$(D, T)$
2: $p(y = 1|x) \leftarrow$ V-SVM$(V, D, T)$

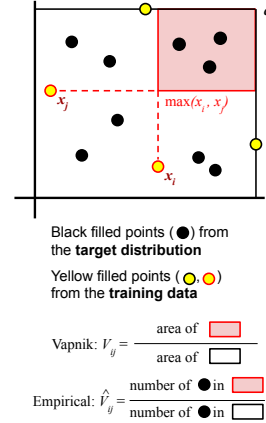

Black filled points (●) from the **target distribution**

Yellow filled points (🟡, ⊙) from the **training data**

Vapnik: $V_{ij} = \dfrac{\text{area of } \blacksquare}{\text{area of } \square}$

Empirical: $\hat{V}_{ij} = \dfrac{\text{number of ● in } \blacksquare}{\text{number of ● in } \square}$

Figure 3: The algorithm for computing the V-matrix and visualization of the steps. The red-border points define the lower-left vertex of the red rectangle. Two other training data-points (shown with black borders) give $c$ which gives the domain boundary of the data. For [17], $V(i, j)$ corresponds to the probability of a point drawn from the uniform distribution falling in the red region. Our empirical $\hat{V}(i, j)$ estimates the *target* probability of falling in the red region.

---
**Algorithm 2** Covariate Shift Classification 2
---
**Input:** $D = (x_1, y_1), (x_2, y_2), \cdots, (x_N, y_N),$
$\qquad\qquad T = t_1, t_2, \cdots, t_M.$
$\qquad\quad$ All $x_i$ and $t_q$ are of dimension $n$.
**Output:** $p(y = 1|x)$
1: **procedure** COMPUTEVADDITIVE($D, T$)
2: $\quad$ $V \leftarrow 0^{N \times N}$
3: $\quad$ **for** $i, j \leq N$ **do**
4: $\qquad$ **for** $q \leq M$ **do**
5: $\qquad\quad$ **for** $l \leq n$ **do**
6: $\qquad\qquad$ **if** $\max(x_i^{(l)}, x_i^{(l)}) \leq t_q^{(l)}$ **then**
7: $\qquad\qquad\quad$ $V_{ij} \leftarrow V_{ij} + \frac{1}{l}$
8: $\qquad$ $V_{ij} \leftarrow \frac{V_{ij}}{M}$
$\quad$ **return** V
1: $V \leftarrow$ COMPUTEVADDITIVE($D, T$)
2: $p(y = 1|x) \leftarrow$ V-SVM($V, D, T$)
---

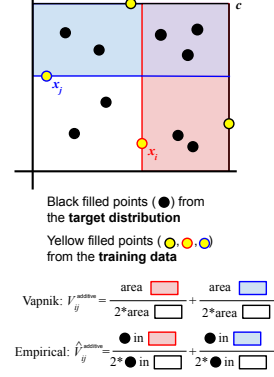

Figure 4: The algorithm for computing the *additive* V-matrix used for higher dimensions and visualization of the steps. Here we take the maximum along dimension $l$, and count the points whose $l$ dimension is larger.

## 3.1 Empirical V-Matrix using the target distribution

Given i.i.d. test data $t_1, t_2, \cdots, t_M$, the empirical cumulative distribution of the target distribution is given by

$$P_M^t(x) = \frac{1}{M} \sum_{q=1}^{M} \theta(x - t_q) \tag{9}$$

Using $P_M^t(x)$ from Eq. (9) for $\mu(x)$ in Eq. (8) gives us an empirical $V(i, j)$ which we denote by $\hat{V}(i, j)$

$$\hat{V}(i, j) = \frac{1}{M} \sum_{q=1}^{M} \theta(t_q - x_i)\theta(t_q - x_j)\phi(t_q). \tag{10}$$

which can be rewritten as $\hat{V}(i, j) = \frac{1}{M} \sum_{q=1}^{M} \phi(t_q) \prod_{k=1}^{n} \mathcal{I}_{t_q^k \geq max(x_i^k, x_j^k)}$. For $\phi(\cdot) = 1$, this is given by

$$\hat{V}(i, j) = \frac{1}{M} \sum_{q=1}^{M} \prod_{k=1}^{n} \mathcal{I}_{t_q^k \geq \max(x_i^k, x_j^k)} \tag{11}$$

where $\mathcal{I}$ denotes the indicator function.

As $\hat{V}(i, j)$ above is a function of test data, the loss function $L(f)$ becomes dependent on test data under this formulation (see Equations 4 and 7).

In Lemma 3.1 below, we show that $\hat{V}(i, j)$ in Eq. (11) is the minimum variance unbiased estimator of the *true V*.

**Lemma 3.1.** *If $\mu^{target}(x) = \prod_{k=1}^{n} \mu_k^{target}(x^k)$ denotes the true cumulative distribution function of the target distribution, then $\hat{V}(i, j)$ in Equation (11) is the minimum variance unbiased estimator (MVUE) of true $V^{true}(i, j)$, where $V^{true}(i, j)$ is obtained using Lemma 2.1.*

Theorem 3.2 stated below is about the rate of convergence of the loss calculated using empirical $V$ to the true loss in terms of the number of target points $M$ for $n = 1$.

**Theorem 3.2.** *Let $\rho^2(V) = \frac{1}{N^2} \sum_{i,j=1}^{N} l_i l_j V(i, j)$, where $l_i = (f(x_i) - y_i)$ and $l_j = (f(x_j) - y_j)$ then for $n = 1$, $|\rho^2(V^{true}) - \rho^2(\hat{V})| \leq \sqrt{\frac{\log M}{M}} \sum_{i,j=1}^{N} \frac{|l_i l_j|}{N^2}$ with probability $\geq 1 - \frac{2}{M^2}$.*

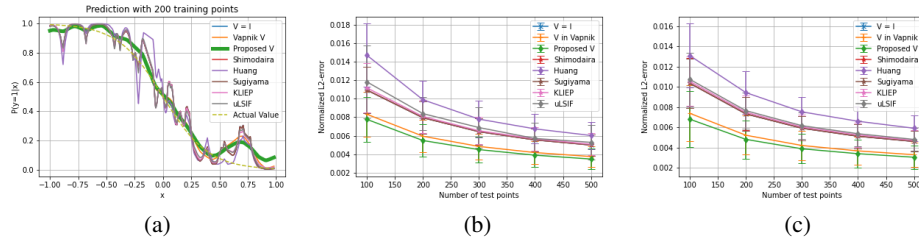

Figure 5: (a) Robustness of methods with 200 training points. (b) Normalized $L2$-error in Experiment 2. (c) Normalized $L2$-error in Experiment 2 using a different target sample.

For more dimensions, we have the following Theorem 3.3.

**Theorem 3.3.** *Given $V^{true}$ and $\hat{V}$, and assuming bounded $|l_i l_j|$ for any $i$ and $j$, the lower bound on the probability that the empirical loss function is within $\delta'$ of the true loss function for a given $\delta > 0$ is given by* $\Pr\left(\left|\rho^2(V^{true}) - \rho^2(\hat{V})\right| \leq \delta'\right) \geq 1 - N(N+1)e^{-2M\delta^2}$, *where* $\delta' = \delta \max_{i,j} |l_i l_j|$. *This suggests that $M > c\frac{\log(N)}{\delta^2}$ with $c > 1$ leads to convergence in probability for any $\delta > 0$.*

**Note:** The proofs of Lemma 3.1, Theorems 3.2 and 3.3 are given in the Supplementary Material.

## 3.2 Algorithm

Our algorithm uses a sample from the target distribution, $T$ to calculate the $V$-matrix. The worst case complexity of calculating the $V$-matrix is $O(N^2 M)$. The full complexity of learning will depend on the choice of learning algorithm. We describe these algorithms for the $V$-matrix proposed in Eq. (11) for the $V$-SVM learning algorithm.

This framework gives a loss function in Eq. (7), so it can be easily adapted to different choices of learning models like decision tree based gradient boosting [4]. See the supplementary section for additional details on using the V-matrix framework in the context of gradient boosting.

**Algorithmic Remark:** In higher dimensions the $\hat{V}$-matrix stated in Eq. 11 can be ill-conditioned. [17] suggests using an additive version of the $V$-matrix. The derivation this version considers the empirical CDF for each dimension separately, minimizing the sum of the $L2$ errors (for details refer Equation 55 in [17]). Thus, Eq. 11 instead becomes $\hat{V}^{additive}(i,j) = \frac{1}{nM}\sum_{q=1}^{M}\sum_{k=1}^{n}\mathcal{I}_{t_q^k \geq max(x_i^k, x_j^k)}$. This is described in Algorithm 2. We use $\hat{V}^{additive}$ for some of our experiments on real datasets.

## 3.3 Experiments on Synthetic Datasets

We conducted multiple experiments to show the applicability and effectiveness of the *proposed* $\hat{V}$-matrix given by Eq. (11). In both the experiments below, a gaussian KDE with bandwidth $h = 2.0$ is chosen for [11, 14, 19]. Further $\tau = 0.5$ for [14]. For Huang et al. [5] method, $B = 1000, \epsilon = \frac{\sqrt{N}-1}{\sqrt{N}}, \sigma = 0.1$, where $N$ is the size of training set. KLIEP and uLSIF are evaluated using the matlab code on the authors' website [15, 6].

The training data comprises $\{(x_i, y_i)\}_{i=1}^{N}$, where $x_i \in [-1,1]$ and $y_i \in \{0,1\}$ such that $p(y_i = 1|x_i) = \frac{1}{1+e^{5x_i}}$. The training data $\{x_i\}_{i=1}^{N}$ is generated using distribution $p(x) \sim \text{Uniform}[-1,1]$ and the test data $\{t_q\}_{q=1}^{M}$ is generated using distribution $q(t) \sim \text{Uniform}[0,1]$ with probability 0.3 and Uniform$[-1,0]$ with probability 0.7.

In Experiments 1 and 2 below, a SVM is used with a square rooted Gaussian kernel with kernel width = 1 and regularization coefficient $\gamma = 0.1$.

**Experiment 1: Robustness**
Figure 5a shows the stability of the predicted $p(y = 1|x)$ for different methods *on a single run*. $N = 200$ training points and $M = 1000$ testing points were used.

We demonstrate that, like the $V$-matrix derived in [17] (given in Example 2.2), the proposed $\hat{V}$-matrix in Equation 11 is stable in its prediction of conditional probability. In contrast, the predictions by unweighted $L2$-loss ($V = I$) and the weighted $L2$-loss using Shimodaira et al. [11], Huang et al. [5], Sugiyama et al. [14, 15], and Kanamori et al. [6] are less smooth, indicating instability.

**Experiment 2: Probability Prediction Error**
Figure 5b compares the normalized $L2$-error of the predicted conditional probability function for various methods. The V-matrix framework is designed to minimize this error, so the improvement is not surprising. Our empirical V-matrix further improves this error because it takes into account the differences between the training and target distributions. Figure 5c instead calculates this V-matrix on a *different* set of 500 samples from the test distribution. Both Figures 5b and 5c are averaged over 50 trials.

*Remark:* The error curves for Sugiyama et al. [14], KLIEP [15], uLSIF [6] and Shimodaira et al. [11] methods in Figures 5b and 5c are overlapping as the mean errors are close.

## 3.4    Experiments on Real Datasets

These experiments use typical datasets from previous papers [5, 6] and synthetically generated covariate shift, as is standard in [5, 14, 15, 6]. [11, 14] represent different regularizations of the ratio of KDE densities and perform similarly, so we only show [11]. Further, the performance obtained for uLSIF [6] and RuLSIF [19] are similar, so we have only included uLSIF. Results given by [5] may not be optimally tuned, as finding the best values for hyper-parameters is difficult. In all the experiments below, we normalize all feature vectors to be in $[0, 1]^n$ ($n$ is the dimension of feature vector).

**Experiment 3: Replicating [6]**
In this experiment, we bias the datasets by following the method used in [6]. We randomly choose $c$ from 1 to $n$ and fix it for each trial. A sample $x_k$ is chosen randomly from the dataset and is accepted in the test (target) set with probability $\min(1, 4(x_k^c)^2)$. The sample $x_k$ is removed from the dataset whether or not it was accepted into the test set. We continue until we have 500 test samples. A training set of size 100 is chosen uniformly at random from the remaining samples in the dataset. The mean performance error over 100 trials for twonorm and ringnorm datasets [9] is provided in Table 1. For the ringnorm dataset, 5 features were randomly chosen in each trial for classification. We used additive version of the V-matrix when testing our method.

In Table 1, we observe a consistent good performance by our method for both the datasets, outperforming other methods on twonorm data. KLIEP and uLSIF do not improve on the twonorm data but perform the best on the ringnorm data. The cancer, diabetes and banknote datasets did not have enough samples to generate 500 test points with the biasing technique described here, so we restricted this experiment to the datasets used in [6].

**Experiment 4: Single Feature Bias**
This experiment is a generalization of the one used in [5]. 100 training points are biased by first randomly selecting a single feature. We then decide to bias the feature up or down down. If biasing up, we increase the probability of selecting a datapoint with a feature value larger than the median by a factor of 4. If biasing down, we decrease its probability by a factor of 4. Each dataset biasing was performed 100 times. Here we use the additive version of the empirical V-matrix.

This biasing scheme is challenging for the previous methods – none are able to achieve significant improvement over the unweighted classifier for the cancer, diabetes, banknote, and ringnorm datasets. Our method improves performance in the banknote and ringnorm datasets. Further, our method is stable in that it never significantly exceeds the error of an unweighted classifier (see cancer and twonorm in Table 1). [6] occasionally performs significantly worse than an unweighted classifier.

**Experiment 5: Multiple Feature Bias**
This experiment is similar to Experiment 4, except that it biases the $L2$ norm of the feature vector instead of a specific feature. The biasing scheme for the norm is the same as given for a single feature in Experiment 4. That is, we bias the selection of datapoints with larger than median $L2$ norm up or down by a factor of 4. Because there is less variation between the datasets generated by this biasing method, the mean performance error is given over 50 trials. As with Experiment 4, some datasets

paired with this biasing scheme are too difficult for any of the methods tested. Our method is the only method to improve significantly in the banknote and ringnorm datasets. The twonorm dataset is the only case where our method underperforms an unweighted classifier, and it does so with a large variance.

| Experiment | Dataset | Our Method | Shimodaira [11] | Huang [5] | KLIEP [15] | uLSIF [6] |
|---|---|---|---|---|---|---|
| Replicating [6] | twonorm (20) | **0.935(0.118)** | 1.001(0.008) | 0.939(0.236) | 1.000(0.159) | 1.003(0.061) |
| | ringnorm (20) | 0.967(0.084) | 1.000(0.002) | 0.985(0.300) | 0.873(.244) | 0.909(0.241) |
| Single feature bias | cancer (9) | 1.072(0.122) | 1.029(0.111) | 1.139(0.189) | 1.019(0.107) | 1.779(2.444) |
| | diabetes (8) | **0.994(0.054)** | 1.002(0.017) | 1.059(0.080) | 1.001(0.034) | 1.019(0.069) |
| | banknote (4) | **0.951(0.242)** | 0.995(0.021) | 1.006(0.320) | 1.015(0.250) | 1.155(0.542) |
| | ringnorm (20) | **0.905(0.086)** | 1.000(0.004) | 1.396(0.164) | 1.046(0.059) | 1.018(0.047) |
| | twonorm (20) | 1.187(0.200) | 0.998(0.013) | 0.952(0.186) | 1.063(0.159) | 1.001(0.072) |
| Multiple feature bias | cancer (9) | 1.075(0.115) | 0.938(0.736) | 1.097(0.177) | 1.015(0.057) | 1.370(1.813) |
| | diabetes (8) | 1.006(0.039) | 1.020(0.137) | 1.039(0.082) | 0.997(0.014) | 0.999(0.019) |
| | banknote (4) | **0.919(0.216)** | 1.005(0.089) | 1.032(0.248) | 1.008(0.110) | 1.208(0.496) |
| | ringnorm (20) | **0.923(0.081)** | 0.978(0.032) | 1.334(0.124) | 1.032(0.060) | 1.015(0.032) |
| | twonorm (20) | 1.247(0.277) | 0.987(0.084) | 0.959(0.203) | 1.054(0.183) | 0.994(0.034) |

Table 1: A comparison of mean(standard deviation) error for multiple methods on real datasets. Results are normalized so that $1.0$ indicates equal performance to a unweighted classifier. We have colored cells blue if they represent an improvement over the unweighted case, and red if they perform worse or have large instability (mean + std > 1.2). The number of input dimensions in each dataset is given in parentheses.

# 4 Conclusion

Our method makes two important contributions towards more effective ways to handle covariate shift. First, it removes the need to tune parameters like kernel bandwidth by switching to a CDF-based framework. Second, in addition to having comparable or superior performance in classification tasks, our framework is significantly more robust in its predictions of conditional probability of labels.

Though we focused on the pairwise classification problem in this paper, our covariate shift method can also be extended to multiclassification by using one-hot encoding for labels and to regression by using the regression version of the Fredholm integral equation (see details in [17] for these settings). Further, the V-matrix framework is not limited to SVMs and can be applied to other machine learning algorithms using the loss function described in Eq. (7). For instance, we derive the closed form solution for the gradient boosting algorithm [4], and give this loss function in the supplementary material. These loss functions can be minimized by standard gradient descent techniques.

A limitation is our method's inability to handle large dimensional data. While using the additive version of the V-matrix provides a fix, future work may explore forms of dimension reduction that allow the use of the original multiplicative V-matrix, which may further improve performance.

## Broader Impact

Machine learning is limited by the availability and quality of data. In many circumstances we may not have access to labeled data from our target distribution. Improving methods for covariate shift will help us extend the impact of the data we do have.

Stably predicting a conditional probability distribution has an application that has recently gained notorious prominence. In the presence of a pandemic, one may wish to predict the death-rate to calculate the expected toll on society. This translates exactly to predicting the conditional probability of dying given demographic information. Data that has been gathered often has a large sampling bias, since a persons risk profile may affect their willingness to leave home and participate in a study, and their age may affect their availability for an online survey. A stable method for covariate shift in this situation can be a critical part of ensuring officials have accurate statistics when making decisions.

One potential negative impact of the work would be if it were to be misunderstood and misused, yielding incorrect results. This is a concern in all areas of data science, so it is important that the conditions for appropriate use be well understood.

## Acknowledgments and Disclosure of Funding

This work is supported by supported by the National Science Foundation Graduate Research Fellowship under Grant No. DGE-1745301, NSF Grant No. CCF-1717884 and The Carver Mead New Adventure Fund.

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
