[Supplementary Material 1]

# Supplementary Material

## Proofs

### Proof of Lemma 2.1

*Proof.*

$$V(i,j) = \int_{\mathbb{R}^n} \theta(x - x_i)\theta(x - x_j)\phi(x)d\left(\prod_{k=1}^{n} \mu_k(x^k)\right).$$

$$V(i,j) = \prod_{k=1}^{n} \int_{\mathbb{R}} \theta(x^k - x_i^k)\theta(x^k - x_j^k)d\mu_k(x^k) = \prod_{k=1}^{n} \int_{max(x_i^{k-}, x_j^{k-})}^{\infty} d\mu_k(x^k)$$

$$= \prod_{k=1}^{n} \left[1 - \mu_k(max(x_i^{k-}, x_j^{k-}))\right].$$

$\square$

### Proof of Lemma 3.1

*Proof.* Consider a Bernoulli trial $Z(t)$, with

$$Z(t) = \prod_{k=1}^{n} \mathcal{I}_{t^k \geq max(x_i^k, x_j^k)}.$$

Therefore

$$p(Z(t) = 1) = \prod_{k=1}^{n} \left[1 - \mu_k^{target}(max(x_i^{k-}, x_j^{k-}))\right].$$

Also note from Lemma 2.1,

$$V^{true}(i,j) = \prod_{k=1}^{n} \left[1 - \mu_k^{target}(max(x_i^{k-}, x_j^{k-}))\right].$$

Hence, we have $V^{true}(i,j) = p(Z(t) = 1)$. $\hat{V}(i,j)$ can be rewritten as $\hat{V}(i,j) = \frac{1}{M}\sum_{q=1}^{M} Z(t_q)$. Therefore, $\hat{V}(i,j)$ denotes the sample mean of the Bernoulli trial $Z(t)$ with samples $Z(t_1), Z(t_2), \cdots, Z(t_m)$. We know that sample mean of a Bernoulli trial is unbiased and a sufficient statistic for $p(Z(t) = 1)$. Hence, $\hat{V}(i,j)$ is the minimum variance unbiased estimator of $p(Z(t) = 1)$ [Duda and Hart, 1973].

$\square$

### Proof of Theorem 3.2

*Proof.* For ease of notation, we prove the theorem assuming continuous $\mu^{target}(.)$ (check footnote[1])

For convenience assume $m_{ij} = max(x_i, x_j)$. Further we know that

$$V^{true}(i,j) = 1 - \mu^{target}(m_{ij}),$$

$$\hat{V}(i,j) = \frac{1}{M}\sum_{q=1}^{M}\mathcal{I}_{t_q \geq m_{ij}} = 1 - \frac{1}{M}\sum_{q=1}^{M}\mathcal{I}_{t_q < m_{ij}}$$

We define

$$D_{KS} \triangleq sup_x \left| \frac{1}{M}\sum_{q=1}^{M}\mathcal{I}_{t_q < x} - \mu^{target}(x) \right|$$

$D_{KS}$ is popularly known as Kolmogorov-Smirnov distance [Kolmogorov, 1933, Smirnov, 1948]. Therefore, we have

$$\left| V^{true}(i,j) - \hat{V}(i,j) \right| = \left| \frac{1}{M}\sum_{q=1}^{M}\mathcal{I}_{t_q < m_{ij}} - \mu^{target}(m_{ij}) \right|$$

$$\leq D_{KS}$$

$$\left| \rho^2(V^{true}) - \rho^2(\hat{V}) \right| = \left| \frac{1}{N^2}\sum_{i,j=1}^{N} l_i l_j \left[ V^{true}(i,j) - \hat{V}(i,j) \right] \right|$$

$$\leq \frac{1}{N^2}\sum_{i,j=1}^{N} |l_i l_j| \left| V^{true}(i,j) - \hat{V}(i,j) \right|$$

$$\leq \frac{1}{N^2}D_{KS}\sum_{i,j=1}^{N} |l_i l_j|$$

Now using Massart's Inequality [Dudley, 2014], $Pr(D_{KS} > \epsilon) < 2e^{-2M\epsilon^2}$. Therefore the statement of theorem follows by choosing $\epsilon = \sqrt{\frac{\log M}{M}}$. □

**Proof of Theorem 3.3**

*Proof.* We first note that

$$\left| \rho^2(V^{true}) - \rho^2(\hat{V}) \right| \leq \frac{1}{N^2}\sum_{i,j} |l_i l_j| \left| V^{true}(i,j) - \hat{V}(i,j) \right|. \tag{1}$$

We prove this theorem by giving a bound for $\frac{1}{N^2}\sum_{i,j} \left| V^{true}(i,j) - \hat{V}(i,j) \right|$. From the proof of Lemma 3.1, $\hat{V}(i,j)$ denotes the sample mean of i.i.d. Bernoulli random variables with $V^{true}(i,j)$ being the expected value. Hence, by the Hoeffding inequality

$$\Pr(\left| V^{true}(i,j) - \hat{V}(i,j) \right| > \delta) < 2e^{-2M\delta^2}$$

Now, we can use the union bound and the fact that $|V^{true}(i,j) - \hat{V}(i,j)| = |V^{true}(j,i) - \hat{V}(j,i)|$ to get

$$\Pr(\frac{1}{N^2}\sum_{i,j}\left|V^{true}(i,j) - \hat{V}(i,j)\right| \leq \delta)$$

$$\geq 1 - 2(\frac{N(N-1)}{2} + N)e^{-2M\delta^2}$$

$$= 1 - N(N+1)e^{-2M\delta^2}$$

Now letting $\delta' = \delta \max_{i,j}|l_i l_j|$, we use inequality (1) to obtain the required result. $\qquad\square$

## Gradient Boosting with $V$ matrix

In the gradient boosting framework introduced by Friedman [Friedman, 2001], the estimate $\hat{y}_i$ of $y_i$ is mathematically modeled as follows:

$$\hat{y}_i = \sum_{k=1}^{K} f_k(x_i), \quad f_k \in \mathcal{F} \tag{2}$$

where $K$ is the number of trees, $f$ is a function the functional space $\mathcal{F}$ and $\mathcal{F}$ is the set all possible Classification and Regression Trees (CARTs). The objective function to be optimized is given by

$$obj = \sum_{i=1}^{n} l(y_i, \hat{y}_i) + \sum_{k=1}^{K} \Omega(f_k) \tag{3}$$

However, in this loss function the samples $x_i$ are assumed to be independent. In a realistic scenario, samples may not be independent, for example, the samples may be derived from different races and genders giving rise to implicit bias in the training dataset. To encounter this, we can have a loss function that takes into account the associations between different loss function. This is tackled by using a V-matrix which also measures the distance between the sample points. Formally, the objective or loss function is given as follows:

$$obj = \sum_{i=1}^{n}\sum_{j=1}^{n} l(y_i, y_j, \hat{y}_i, \hat{y}_j, V_{ij}) + \sum_{k=1}^{K} \Omega(f_k) \tag{4}$$

Note we use $V_{ij}$ to denote $V(i,j)$ here. The proof below deals with a special case when $l(y_i, y_j, \hat{y}_i, \hat{y}_j, V_{ij}) = (y_i - \hat{y}_i)(y_j - \hat{y}_j)V_{ij}$.

## Proof

$$obj = \sum_{i=1}^{n}\sum_{j=1}^{n}(y_i - \hat{y}_i)(y_j - \hat{y}_j)V_{ij} + \sum_{k=1}^{K} \Omega(f_k) \tag{5}$$

$$\hat{y}_i^{(t)} = \sum_{k=1}^{t} f_k(x_i) = \hat{y}_i^{(t-1)} + f_t(x_i) \tag{6}$$

$obj^{(t)}$

$$= \sum_{i,j=1}^{n} (y_i - [\hat{y}_i^{(t-1)} + f_t(x_i)])(y_j - [\hat{y}_j^{(t-1)} + f_t(x_j)])V_{ij}$$

$$+ \sum_{k=1}^{t} \Omega(f_k)$$

$$= obj^{(t-1)} + \sum_{i,j=1}^{n} f_t(x_j)(\hat{y}_i^{(t-1)} - y_i)V_{ij} + \tag{7}$$

$$\sum_{i,j=1}^{n} [f_t(x_i)(\hat{y}_j^{(t-1)} - y_j) + f_t(x_i)f_t(x_j)]V_{ij} + \Omega(f_t)$$

$$= \sum_{i,j=1}^{n} [f_t(x_j)(\hat{y}_i^{(t-1)} - y_i) + f_t(x_i)(\hat{y}_j^{(t-1)} - y_j)]V_{ij}$$

$$+ \sum_{i,j=1}^{n} f_t(x_i)f_t(x_j)V_{ij} + \Omega(f_t) + constant$$

$f_t(x)$ and $\Omega(f_t)$ are defined as follows:

$$f_t(x) = w_{q(x)}, \ w \in \mathbb{R}^T, \ q : \mathbb{R}^d \to \{1, 2, \cdots, T\}. \tag{8}$$

where $w$ is the vector of scores on leaves, $q$ is a function assigning each data point to the corresponding leaf, and $T$ is the number of leaves. The complexity $\Omega(f_t)$ is given by

$$\Omega(f_t) = \gamma T + \frac{1}{2}\lambda \sum_{j=1}^{T} w_j^2 \tag{9}$$

Let $g_i = \hat{y}_i^{(t-1)} - y_i$, and $I_j$ be the set of all $x_i$ that belong to leaf $j$, i.e. $I_j = \{i|q(x_i) = j\}$, then

$$
\begin{aligned}
obj^{(t)} &= \sum_{i,j=1}^{n} [f_t(x_j)g_i + f_t(x_i)g_j + f_t(x_i)f_t(x_j)]V_{ij} \\
&\quad + \Omega(f_t) + constant \\
&= \sum_{i,j=1}^{n} [w_{q(x_j)}g_i + w_{q(x_i)}g_j + w_{q(x_i)}w_{q(x_j)}]V_{ij} \\
&\quad + \Omega(f_t) + constant \\
&= \sum_{k=1}^{T} w_k [\sum_{i=1}^{n}\sum_{j \in I_k} g_i V_{ij} + \sum_{i \in I_k}\sum_{j=1}^{n} g_j V_{ij}] \\
&\quad + \sum_{l=1}^{T}\sum_{m=1}^{T} w_l w_m \sum_{i \in I_l}\sum_{j \in I_m} V_{ij} + \frac{1}{2}\lambda \sum_{k=1}^{T} w_k^2 \\
&\quad + \gamma T + constant \\
&= \sum_{k=1}^{T} w_k [A_k + B_k] + \sum_{l=1}^{T}\sum_{m=1}^{T} w_l w_m C_{lm} \\
&\quad + \frac{1}{2}\lambda \sum_{k=1}^{T} w_k^2 + \gamma T + constant
\end{aligned}
\tag{10}
$$

where

$$
A_k = \sum_{i=1}^{n}\sum_{j \in I_k} g_i V_{ij} = \sum_{i=1}^{n} g_i \sum_{j \in I_k} V_{ij}
\tag{11}
$$

$$
B_k = \sum_{i \in I_k}\sum_{j=1}^{n} g_j V_{ij} = \sum_{j=1}^{n} g_j \sum_{i \in I_k} V_{ij}
\tag{12}
$$

$$
C_{lm} = \sum_{i \in I_l}\sum_{j \in I_m} V_{ij}
\tag{13}
$$

Taking partial derivative of $obj^{(t)}$ with respect to $w_i$ gives us

$$
A_i + B_i + \sum_{j=1}^{T} [w_j (C_{ij} + C_{ji})] + \lambda w_i = 0 \ \forall i \in \{1, 2, \cdots, T\}.
\tag{14}
$$

Eq. (14) can be rewritten as:

$$
Dw^* = U.
\tag{15}
$$

where $w^*$ is the optimal $w$, $D$ is a $T \times T$ matrix with

$$
D_{ij} = C_{ij} + C_{ji}, \ j \neq i
$$

$$D_{ii} = 2C_{ii} + \lambda.$$

or in other words

$$D = C + C^T + \lambda I, \ I : identity\ matrix$$

Also note that $D^T = D$. $U$ is a $T \times 1$ vector with

$$U_i = -(A_i + B_i).$$

If $D$ is invertible, then

$$w^* = D^{-1}U. \tag{16}$$

The $obj^{(t)}$ function from Eq. (10) can be rewritten as:

$$obj^{(t)} = -w^T U + w^T C^T w + \frac{1}{2}\lambda w^T w. \tag{17}$$

For $w = w^*$, $obj^{(t)}$ denoted by $obj^*$ is given by:

$$\begin{aligned}
obj^* &= -U^T (D^{-1})^T U + U^T (D^{-1})^T C^T D^{-1} U \\
&\quad + \frac{1}{2}\lambda U^T (D^{-1})^T D^{-1} U + \gamma T + constant \\
&= -U^T (D^T)^{-1} U + U^T (D^T)^{-1} C^T D^{-1} U \\
&\quad + \frac{1}{2}\lambda U^T (D^T)^{-1} D^{-1} U + \gamma T + constant \\
&= -U^T D^{-1} U + U^T D^{-1} C^T D^{-1} U \\
&\quad + \frac{1}{2}\lambda U^T D^{-1} D^{-1} U + \gamma T + constant
\end{aligned} \tag{18}$$

When $V_{ij} = V_{ji}$, $obj^*$ in Eq. (18) above can be further simplified. Note that, here $C^T = C$ which implies

$$C^T = \frac{1}{2}[D - \lambda I]. \tag{19}$$

Plugging $C^T$ from (19) above in Eq. (18), we get

$$obj^* = -\frac{1}{2}U^T D^{-1} U + \gamma T + constant. \tag{20}$$

Eqs. (18) and (20) provide a metric to evaluate the goodness of the $t$-th tree in the gradient boosting algorithm.

**Note:** $V_{ij} = V_{ji}$ also implies that $A = B$.

## Footnotes

[1] For cumulative distribution functions that are not left continuous, we can replace $\mu^{target}(x)$ by $\mu^{target}(x^-)$ at points of discontinuity.

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

[Supplementary Material 2 · READ_ME.pdf]

# Experiment 1

**Stability.ipynb** is used to generate the robustness (stability) plot given in Figures 1c and 4a. The file **TX.mat** generated by the code will need to be loaded to the matlab codes of KLIEP and uLSIF for obtaining the importance weights for KLIEP and uLSIF methods. **X (training data)** and **T (testing data)** are the variables stored by **TX.mat.** Once the importance weights are obtained for KLIEP and uLSIF, they can be loaded back into Stability.ipynb and be used to obtain predictions for diagonal V matrices corresponding to KLIEP and uLSIF methods by running the **test_methods_review()** function.

# Experiment 2

We provide **figure4b.ipynb** and **figure4c.ipynb** to generate Figures 4b and 4c. In order to run these files for KLIEP and uLSIF methods (which are implemented in MATLAB), we store the synthetic data in **data.mat** and **data_2.mat** for Figures 4b and 4c respectively. These files are then used to generate importance weights for KLIEP and uLSIF methods using **KLIEP.m** and **uLSIF.m.** The importance weights are stored in **data_final.mat** and **data_final_2.mat** for Figures 4b and 4c respectively. For convenience, we have provided all these files in the folder **dataset_files(mat,pickle).** Further, we have also provided the matlab functions **test.m** and **test_2.m** used to generate **data_final.mat** and **data_final_2.mat** respectively.

# Experiment 3

**Experiment_3_ringnorm.ipynb** and **Experiment_3_twonorm.ipynb** are used in this experiment. The necessary mat and pickle files with datasets obtained by the biasing scheme described in experiment 3 are provided in the folder **dataset_files(mat,pickle).** We have also provided the python notebook **Dataset_Generator_Experiment_3.ipynb** which was used to bias these datasets. We also include the **GetVmatrices_ringnorm.m** and **GetVmatrices_twonorm.m** used to generate **matlab_ringnorm.mat** and **matlab_twonorm.mat** that have importance weights for KLIEP and uLSIF methods for ringnorm and twonorm datasets respectively.

# Experiments 4 and 5
# Generating Datasets

Running these files requires access to the datasets. You will need to update the paths for your setup, but these datasets are given in the supplementary material.

**Dataset_Generator_SingleFeature.ipynb** generates training/testing splits by selecting training datapoints based on one feature as described in the paper. It outputs the files **single_datasets.mat** and **single_datasets.p** which contain the preprocessed data as well as lists of indices for training and testing.

**Dataset_Generator_Norm.ipynb** generates training/testing splits by selecting training datapoints based on their norm as described in the paper. It outputs the files

**multi_datasets.mat** and **multi_datasets.p** which contain the preprocessed data as well as lists of indices for training and testing.

The outputs from these dataset generators are in the **MultiFeatureBias** and **SingleFeatureBias** folders.

## Running KLIEP and uLSIF in Matlab

These algorithms are run using their original code in MATLAB. We provide the original implementations in our folder.

**Get_V_Matrices.m** is run to generate the weightings for each trial. To run this, you must load **single_datasets.mat** or **multi_datasets.mat** into your environment. The code to do this is in the file, but the path may need to be updated depending on your setup. The code must also have access to the matlab implementation of the KLIEP and uLSIF method, provided in the folder. The environment should be saved in **matlab_single.mat** or **matlab_multifeature.mat** depending on whether we are doing the single-feature biasing or the norm-biasing.

## Running the Experiment

**CovariateShiftExperiments.ipynb** runs the experiment. To run this you must have the necessary .p and .mat files for the experiment.

For the single feature experiment you must have **single_datasets.p**, which contains the datasets and train-test splits. You must also have **matlab_single.mat** which contains the weightings for KLIEP and uLSIF. Both of these are in the **SingeFeatureBias** folder.

For the norm experiment you must have **multi_datasets.p**, which contains the datasets and train-test splits. You must also have **matlab_multifieature.mat** which contains the weightings for KLIEP and uLSIF. Both of these are in the **MultiFeatureBias** folder. Again, the paths may need to be updated based on your setup.

To choose the norm-biased or feature-biased experiment, uncomment the code that loads the appropriate files. The comments label each section of code.

To run the experiment for a specific dataset, modify the line:
```
dataset = 'ringnorm'
```
To be `'cancer', 'ringnorm', 'twonorm', 'diabetes', 'banknote'.`

Note: The parameters for Makoto and Huang's methods are not tuned in these files so you will need to tune those parameters yourself to verify their performance.

The order of the output performance is:
  1. Unweighted

2. Proposed
3. V (Multiplicative)
4. Proposed V (Additive)
5. Makoto
6. Huang
7. uLSIF

The running totals are printed after each train-test split.