[Reviews · NeurIPS 2020]

Review 1

Summary and Contributions: - The authors adapt a recent idea by Vaonik and Izmailov to the covariate shift setting, and provide theory on estimation of the V matrix (one component of the method) on the target task (classification here, specifically SVM, boosting). - Their method considers the estimation of the CDF of the target distribution, which the authors empirically show is more stable in estimation of the conditional probabilities, without much parameter tuning compared to previous work.

Strengths: - I am not an expert on the Vaonik and Izmailov and covariate shift literature, but as far as I know the methodology considered is new. - The main contribution comes from replacing the V-matrix (indep of labels) by equation (11), computed using the target X for the purpose of covariate shift. The authors provide some theory that their estimator is MVUE and also provide some convergence rate. - Empirically they show that their methodology has more robust prediction and is comparable/better on a variety of datasets (vs current approaches), though there are some comments below.

Weaknesses: - I would like to see a bit more experiments, especially regarding its performance, as the performance of the proposed method is not beating other approaches by a wide margin (from looking at pure std, it seems to be overlapping mostly, although the mean is lower, and seems to be more stable on the synthetic example). - I would like to see the result of the existing covariate shift methods with some tuning on the hyperparameters, as the authors proposed the cross val is wasteful of data. Here, they were fixed during experiment, perhaps it will be interesting to see with some tuning, how does the existing baselines compare, especially when authors claim their method is more stable and robust practically, compared to other methods. Even an oracle is ok. - The method proposed only works on SVM and boosting (for classification), though the authors says this can be extended.

Correctness: On the empirical side, perhaps the author can check whether the proposed method beat other methods given one set of problem consistently, using say a signed rank permutation test. This is because I imagine the problem can vary a lot in difficulty, and hence the std are likely to overlap.

Clarity: Overall I follow the paper, and the diagrams were useful.

Relation to Prior Work: Yes, the authors discuss this clearly in section 1, and the authors also clearly states their contribution vs that of Vaonik and Izmailov.

Reproducibility: Yes

Additional Feedback: I have a confusion about what is R in line 171, I cannot seem to find its definition in Theorem 3.3. Minor: Typo in Table 1 caption -------------------------------------------- After rebuttal: Overall I think the methodology is nice, however I am still not convinced about the experimental results (and the scenarios that it is useful on real life applications).


Review 2

Summary and Contributions: For a setting of a covariate shift (different feature distributions but the same conditional distributions for source and target), authors look into approaches that train on S data while reweighing the points to account for the shift. Authors propose to estimate weights without having to estimate the densities (instead, using cdfs). The proposed method is more stable and not hyperparams (for densities) sensitive

Strengths: - Derivations of V matrix and applications to gbdt (but it is in supplemental) - Well written

Weaknesses: - Limited impact. Authors demonstrate experiments with SVMs, compare with old methods for covariance shift (2007-2011) - Small datasets experiments, many results are not significant - Not clear how/if the method will scale to millions of instances, and larger feature dimensions (authors state that it won't scale to larger dimensions)

Correctness: Yes

Clarity: Yes, enough background on all the work they build on

Relation to Prior Work: Yes but authors concentrate only on reweighing scheme to combat covariate shift. Any other modern methods for domain adaptation are not mentioned/not compared against

Reproducibility: Yes

Additional Feedback: Update: I read authors' responce, I am shifting my score to marginally above the reshold. The idea is interesting and the vision is important. I do appreciate hyperparameter free solutions, as everyone does I suppose. I do still feel that this paper is not a clear accept. Applying it to non svm settings would strengthen its appeal immensely, and comparing with more modern covariate shift methods will go a long way. Also I don't find the claim that it gets the best performance in half of the experiments compelling enough. ============= As I said before, the paper is ok but I am struggling to understand how useful it is/ what the impact is 1) Why do use SVMs, they are pretty old, do not work on large data and thousands of features. Does your method extends to any other models) like NN (I assume it should because it is weights for source that you are learning). Why not to test on NNs for example? Or gbdts? 2) All the methods you compare to are really old 2007-20011), why not to compare with new things like https://arxiv.org/abs/1505.07818, https://arxiv.org/pdf/1705.10667.pdf etc (nn based and works even with unlabelled target), or even fine tuning? Minor: Experiments:- Table 1 some rows don't have the best method bolden. Also you seem to bolden even if the results are not statistically significant (twonorm for replicating experiment etc) Line 89 missing space before "In section"Section 2 introduce N number of training data points Algorithm 1 should really go after section 3.2


Review 3

Summary and Contributions: This paper borrows the idea from Vapnik and Izmailov (2019) and proposes to construct a V matrix for solving the covariate shift (CS) problem. The main idea is to replace the uniform measure by the target set measure in Eq.(8) for evaluation. Consistency guarantees are provided, and experiments show that the proposed method can perform better than existing CS techniques. =========== Update =========== I thank the authors for their feedback. Overall, I understand and appreciate the idea of CS without parameter-tuning. However, as other reviewers pointed out, the performance gain is marginal in the experiments, and the applicability of the proposed method is limited in practice.

Strengths: - The idea is simple and easy to understand. - Theorems of consistency.

Weaknesses: - The writing can be improved. - Shortcomings are not discussed sufficiently.

Correctness: Other than the technical inaccuracies of several statements (see below), most contents are correct.

Clarity: - Eq.(1) does not involve the label variable Y (as opposite to L34). In fact, the readers may not be as familiar with the topic so it would be better to define the notations clearly and accurately, especially given that this is the first/introductory equation. - When Fig.1c is first introduced in L57, there is no description of what the problem is. - L39: "A key shortcoming of these methods is that they require estimating the probability density function of both the training and target data using density estimation techniques". This statement is invalid. For example, the [19] mentioned before this is not estimating individual densities. - L42, L184, L214: it seems that this paper confuses KDE with linearly parameterized ratio models. Kernel density estimation is to estimate "density". Most existing methods for covariate shift estimate density "ratio". Even though the parametric form can be similar to KDE, it is NOT KDE. - L78: "reduced L2 − prediction" is badly formatted.

Relation to Prior Work: Well discussed.

Reproducibility: Yes

Additional Feedback: What if V(i,j)=0 for all i,j? Looking at (11), this can happen when every target point t<x for all x, especially for high-dimensional data. About the experiments - The loss has gamma in Eq.(4) as a parameter. Is this parameter sensitive? Since the main point of this work is to eliminate tuning hyperparameter, it would be necessary to see that the proposed method is indeed robust to this parameter to some extent. - The experiment in Sec.3.3: if f is the V-SVM in Sec.2.4, how is the kernel width chosen? - Table 1. Not every row has a bolded method. For the twonorm dataset, the proposed method is noticeably worse than other methods. It would be helpful to discuss why this case failed and when the proposed method will fail in general.


Review 4

Summary and Contributions: The paper introduces a new approach to address covariate shift using the empirical cumulative distribution function of the target data. It build heavily on recent work by Vapnik and Izmailov, is conceptually simple and shown to be effective on simulated and real-world tasks.

Strengths: - Novel, conceptually simple and practically useful approach. - Thorough comparison to existing methods. - Transparent experiments. I much appreciate that the others admit that hyper-parameter tuning for competing methods may be sub-optimal.

Weaknesses: - The paper could trade some technical details against giving intuitive explanations

Correctness: Yes

Clarity: Yes

Relation to Prior Work: The paper builds heavily on the work by Vapnik and Izmailov which is clear throughout. It could be made clearer what the original contributions of this work are.

Reproducibility: Yes

Additional Feedback: I think the title is very non-informative. It does not say what the method does nor how. Abstract "Varying domains and biased datasets can lead to differences between the training and the target distributions, a problem termed covariate shift" - There is a difference between covariance shift and more general distribution shift as the authors point out in to introduction. - Not sure what "varying domains can lead to differences in the training and target distribution" should mean(?) Introduction - Eq. (1) assumes perfect generalization. - Text annotations of Fig 1 are too small but there is a lot of whitespace below the figure. Maybe consider a 4x4 figure arrangement with 1/4 being the caption. [Update] Thanks to the authors for their response. My rating remains.

[Author Response · NeurIPS 2020]

We thank the reviewers for their helpful comments. We first provide individual responses to each reviewer's comments and then provide a general response to all the reviewers.

**Reviewer 1:** Tuning of hyperparameters must be done with cross validation or there is no way to ensure that the tuning is not causing over-fitting. If we tune for maximum accuracy and test on the target data separately the models perform very poorly. Handling this overfitting is why the uLSIF and KLIEP papers were written, the algorithms of which rely on finding the right parameters for optimal reweighting using cross-validation.

**Reviewer 3:** The paper presents a novel approach to reweighting data in the covariate shift which unlike other state of the art methods doesn't require parameter tuning and shows promising signs of being more stable. Further work can, and will, expand on how these ideas can be scaled to larger datasets and larger dimensions.

We focus on the SVM because it enables us to provide the most straightforward demonstration of our framework on the real datasets used by other state of the art covariate shift method papers. However, the method is model-independent and can be applied on other datasets where the underlying target function can be better modeled by other model classes like NN or decision tree based GdBt. We also want to point out that SVM might be old but are used heavily in practice when the data is noisy and limited (for example in finance applications). Further, there are many options that can be explored as future work. For example, one could easily apply this method to the last row of a neural network. The NN papers mentioned by the reviewer address domain adaptation while this work addresses the covariate shift problem (though they are qualitatively similar, they differ mathematically).

**Reviewer 4:** We will make the suggested changes to improve the writing. We will correct the typo in Equation 1 and replace $L(f(X), X)$ with $L(f(X), y)$ and define the notation clearly. Fig 1c in L57 is similar to Fig. 4a explained in experiment 3.3 except it uses 100 training points instead of 200. We will make this reference in the text of the paper.

The reviewer correctly points out that [19] doesn't estimate individual densities but directly estimates the weight. We will make the correction. We have tried to distinguish between techniques that estimate individual densities via KDE against directly estimating the ratio of densities in the subsequent paragraph where we mention the methods by Huang et al., KLIEP and uLSIF. We will further clarify those differences. We appreciate you bringing this distinction of terminology. However, it is also important to realize that all of these previous methods still require a *kernel function* with a properly *tuned bandwidth* - which is one of the main shortcoming that our paper addresses.

The shortcomings of our method amount to it needing further work to handle higher dimensions . We explain one way around the instabilities of the CDF in higher dimensions by a heuristic (the "additive" V-matrix), but we believe a rigorous treatment of this will be the subject of additional papers (this could possibly be a reason for worse performance on twonorm dataset).

The paper addresses the hyper-parameter tuning problem that state of the art covariate shift methods face for *finding optimal reweighting.* In the context of our paper, all reweighting is encoded in the V-matrix, a concept introduced by Vapnik et. al that we demonstrate includes all previous methods as a special case. The regularization parameter $\gamma$ in Equation 4 is a learning algorithm parameter, which is not involved in calculating the V-matrix reweighting. Previous methods can be interpreted as calculating diagonal V-matrices for reweighting, and require tuning in order to do this. Similarly, the kernel width for f in V-SVM is a model (learning algorithm) parameter and again should not be confused with parameters that are needed to be necessarily tuned for optimal reweighting. $\gamma = 0.1$ and the kernel width for f is chosen to be 1 for all the methods tested in Experiment 3.3.

**Reviewer 5:** We agree the title is non-informative, but we felt it was necessary to emphasize the main contribution of our work since we wanted to emphasize on two key aspects of our method - robustness and no parameter tuning. Perhaps a better title would be: "Pairwise covariate-shift reweighting from cumulative distribution functions gives robust parameter-free performance"?

**All Reviewers:** In this paper we rethink how we find the optimal reweighting for the covariate shift problems with an emphasis on removing the necessary parameter tuning and increasing the robustness. As we emphasize in our paper, this framework is model-independent. We define a loss function - so any hypothesis (model) class that minimizes a loss function will work. To this end along with giving certain theoretical guarantees for our method, we demonstrate by experiments on synthetic and real datasets that our method gives near similar or improved performance than other methods in most of the cases *without any need for parameter tuning* that the other covariate shift methods heavily rely on for finding the optimal reweighting. *In addition to that, for other covariate shift methods, there is no rigorous way to tune the right parameters. Hence, a method that can avoid the need for parameter tuning without any compromise on the performance (which other methods only achieve if the right set of parameters are tuned) is a novel contribution.* Further, our method shows *stability* in its predicted probability function (shown in our synthetic data experiments), and *consistency* across multiple real datasets. We would like to point out that the state of the art methods we tested, do not provide large improvements from the unweighted case in some experiments. These state of the art methods sometimes do significantly worse than the unweighted cases if the parameters are sub-optimally tuned. The bold face in Table 1 is to emphasize the consistency of our method demonstrating that our method gives the best mean performance for *6 out of 12 experiments* shown in Table 1. While we certainly can add more experiments, we believe the current number (12 real data experiments and 1 synthetic data experiment) is enough to justify our claim. The nature of experiments performed in the paper is an established standard for evaluation in the popular covariate shift papers (uLSIF, Huang et al.).

[Meta-Review · NeurIPS 2020]

The paper is extending the statistical invariants idea of Vapnik and Izmailov to covariate shift setting. The resulting method basically use the V-matrix of the target distribution while using the source labels. The application is not straightforward; moreover, authors perform an analysis and discuss the theoretical properties of the method. Resulting method is also empirically evaluated on some classical datasets. The papers received mix scores from the reviewers. In summary, - All reviewers agreed on the novelty, correctness, clarity, and the theoretical value of the paper. - R#1 and R#4 raised issues on the empirical study stated that the margins are narrow and baselines can be improved with more hyper-parameter search. - R#3 raised issues stating SVMs are pretty old, the method should used NNs. Moreover, it should compare with NN based methods. First of all, I disagree with the R#3 in extending the method to NNs. Extending the proposed method to NN is very non-trivial due to the curse-of-dimensionality and heavy use of RKHS/Kernels in the V-matrix which do not exist in NNs. The remaining issues are clearly not valid reasons for rejecting a paper. Because 1) As long as there is non-trivial theoretical/algorithmic advancements and the empirical study is solid, we do not need state-of-the-art results with large margins. 2) Proposed method does not use hyperparameter search and expecting it from baselines is not fair. 3) Not all methods need to use neural networks, we still need diverse ideas and models in NeurIPS for an healthy field.